# Computational analysis of pathological images enables a better diagnosis of TFE3 Xp11.2 translocation renal cell carcinoma

Jun Cheng [1], Zhi Han [2,3], Rohit Mehra [4], Wei Shao [2], Michael Cheng[2], Qianjin Feng [5], Dong Ni [1✉], Kun Huang [2,3✉], Liang Cheng[6✉] & Jie Zhang [7✉]

TFE3 Xp11.2 translocation renal cell carcinoma (TFE3-RCC) generally progresses more aggressively compared with other RCC subtypes, but it is challenging to diagnose TFE3-RCC by traditional visual inspection of pathological images. In this study, we collect hematoxylin and eosin- stained histopathology whole-slide images of 74 TFE3-RCC cases (the largest cohort to date) and 74 clear cell RCC cases (ccRCC, the most common RCC subtype) with matched gender and tumor grade. An automatic computational pipeline is implemented to extract image features. Comparative study identifies 52 image features with significant differences between TFE3-RCC and ccRCC. Machine learning models are built to distinguish TFE3-RCC from ccRCC. Tests of the classification models on an external validation set reveal high accuracy with areas under ROC curve ranging from 0.842 to 0.894. Our results suggest that automatically derived image features can capture subtle morphological differences between TFE3-RCC and ccRCC and contribute to a potential guideline for TFE3-RCC diagnosis.

[1] National-Regional Key Technology Engineering Laboratory for Medical Ultrasound, Guangdong Key Laboratory for Biomedical Measurements and Ultrasound Imaging, School of Biomedical Engineering, Health Science Center, Shenzhen University, Shenzhen, China. [2] Department of Medicine, Indiana University School of Medicine, Indianapolis, IN, USA. [3] Regenstrief Institute, Indianapolis, IN, USA. [4] Department of Pathology, University of Michigan, Ann Arbor, MI, USA. [5] School of Biomedical Engineering, Southern Medical University, Guangzhou, China. [6] Department of Pathology and Laboratory Medicine, Indiana University School of Medicine, Indianapolis, IN, USA. [7] Department of Medical and Molecular Genetics, Indiana University, Indianapolis, IN, USA. ✉email: nidong@szu.edu.cn; kunhuang@iu.edu; lcheng@iupui.edu; jizhan@iu.edu

Renal cell carcinoma (RCC) consists of multiple heterogeneous subtypes[1,2] and is canonically classified into three major histologic subtypes: clear cell RCC (ccRCC) (~75%), papillary RCC (15–20%), and chromophobe RCC (~5%)[3,4]. In addition to the histopathologically defined subtypes of RCC, the Xp11.2 translocation RCC, a rare subtype associated with TFE3 gene fusion, was first officially recognized in the 2004 WHO renal tumor classification. The TFE3 gene, which is located on chromosome Xp11.2, has various fusion partners[5–7]. Renal cell carcinomas with t(6;11) translocation, harboring a MALAT1-TFEB gene fusion, are far less common.

TFE3 Xp11.2 translocation RCC (TFE3-RCC) is often diagnosed at advanced stage and demonstrates a more invasive clinical course and poorer prognosis than non-Xp11.2 translocation RCC. Significant progress has been achieved by targeted therapies for kidney cancer treatment in recent years[8], in particular VEGF-targeted (sunitinib, sorafenib, etc.) and mTOR-targeted (temsirolimus, everolimus, etc.) therapies that block angiogenic activity[9–11]. During the past few years, there have been many studies investigating the efficacy of targeted therapies for patients with TFE3-RCC[7,12–16]. For instance, Choueiri et al.[14] showed that VEGF-targeted agents demonstrated some efficacy in patients with metastatic TFE3-RCC in a small retrospective review. Improving underdiagnosis of this rare subtype of RCC will facilitate sample curation, improve clinical trial access, and more importantly, contribute to the development of effective therapies for this group of patients.

However, it is quite challenging to distinguish TFE3-RCC from other subtypes based on visual inspections of hematoxylin and eosin (H&E)-stained pathological images. The gross morphology of TFE3-RCC is similar to that of ccRCC[5–7,17]. Microscopically, TFE3-RCC cases often feature epithelioid clear cells arranged in branching, papillary structures with fibrovascular cores and/or a nested architecture. Although these features are suggestive of TFE3-RCC, the spectrum of morphology is quite variable and can overlap with other RCC subtypes such as ccRCC or papillary RCC[1,2]. For instance, some cases in the ccRCC and papillary RCC datasets of The Cancer Genome Atlas (TCGA) project are related to TFE3 or TFEB translocation[18,19].

Due to the difficulty of identifying discernable and robust morphological features in TFE3-RCC, the diagnosis of translocation can be confirmed by dual-color, break-apart fluorescence in situ hybridization. However, it requires additional time to test for this diagnosis, and it is not routinely performed for the RCC patients who are not suspected of TFE3-RCC in the first place. Therefore, there is a high risk that TFE3-RCC is misdiagnosed with other RCC subtypes, which delays appropriate treatments. We want to apply machine learning to digitized H&E-stained pathological images and study whether it can help identify TFE3-RCC unique image features and distinguish TFE3-RCC from the most common RCC subtype, ccRCC.

As digital slide scanners have become more reliable and popular, glass slides have been increasingly digitized into whole-slide images. Recent years have witnessed a growing interest in applying machine learning to H&E-stained pathological images for various tasks including prognosis prediction[20–22], cancer classification[23–26], and genetic status prediction, such as microsatellite instability[27] and gene mutation[28]. Notably, Campanella et al.[23] reported a clinical-grade computational pathology framework that was evaluated on a dataset of 44,732 whole-slide images. Combining image processing techniques and machine-learning models, Yu et al.[26] achieved an area under the curve (AUC) of 0.85 in distinguishing normal from tumor slides and 0.75 in differentiating between lung adenocarcinoma and squamous cell carcinoma slides. These studies demonstrated the efficacy of computational pathology in clinical decision support.

In this study, we collect H&E-stained whole-slide images for 74 TFE3-RCC patients from multiple sources (the largest reported study on TFE3-RCC based on our knowledge) and 74 gender and tumor grade matched ccRCC patients. The aims of the study are (i) to identify distinct, quantitative image features showing significant differences between TFE3-RCC and ccRCC; and (ii) to build and evaluate objective and fully automated classification models based on these features to distinguish TFE3-RCC from ccRCC.

## Results

**Patient characteristics and pathological image analysis workflow.** We collected two whole-slide image datasets: dataset 1 and dataset 2. Dataset 1 was obtained from Indiana University, consisting of 50 TFE3-RCC patients and 50 ccRCC patients with matched gender and tumor grade. Dataset 1 was randomly split into training (80%) and internal validation (20%) sets for five times using five-fold cross-validation. Dataset 2 was obtained from University of Michigan and TCGA. It was used as an external validation set. It contains 24 TFE3-RCC patients and 24 ccRCC patients, also with matched gender and tumor grade. Patient demographical and clinical characteristics of the two datasets are summarized in Table 1.

The analysis workflow is shown in Fig. 1. The H&E-stained slides of the 148 excisional biopsy cases were digitized by a Leica Aperio scanner at ×40 magnification (Fig. 1a). A pathological image analysis pipeline extracted quantitative image features from whole-slide images[21], characterizing the size, staining, shape, and density of cell nuclei (Fig. 1b). To study the associations of the image features with disease subtype (i.e., TFE3-RCC vs ccRCC; Fig. 1c), first the distribution of each image feature was compared between the two subtypes using the Mann–Whitney U test. Then, the image features were combined and four machine-learning models (logistic regression, SVM with linear kernel, SVM with Gaussian kernel, and random forest) were built to classify patients into TFE3-RCC or ccRCC group.

The feature extraction pipeline consisted of three steps: nucleus segmentation, nucleus-level feature extraction, and image-level feature extraction (Fig. 2). First, cell nuclei in whole-slide images were segmented by a hierarchical multilevel thresholding approach[29] (Fig. 2a). Next, for each segmented nucleus, 10 nucleus-level features were calculated (Fig. 2b). Representative image patches of the 10 nucleus-level features are shown in Table 2. Lastly, since each whole-slide image contains millions of cell nuclei, each type of nucleus-level features was dissected into 15 image-level features by combining a 10-bin histogram and 5 distribution statistics (mean, std, skewness, kurtosis, and entropy) (Fig. 2c). The bin centers of the histogram were cluster centroids determined by clustering each type of nucleus-level features sampled from the training set; hence, the histogram features are comparable across patients. The naming rule of the 15 image-level features is shown in Fig. 2c, using the nucleus-level feature (e.g., ratio). In total, we calculated 150 image-level features for

### Table 1 Demographic and tumor characteristics of two whole-slide image datasets.

| Characteristics | Dataset 1: TFE3-RCC/ccRCC | Dataset 2: TFE3-RCC/ccRCC |
|---|---|---|
| No. of patients | 50/50 | 24/24 |
| Gender: Male | 22/22 | 9/9 |
| Gender: Female | 28/28 | 15/15 |
| Fuhrman grade: 2 | 10/10 | 6/6 |
| Fuhrman grade: 3 | 29/29 | 15/15 |
| Fuhrman grade: 4 | 11/11 | 3/3 |

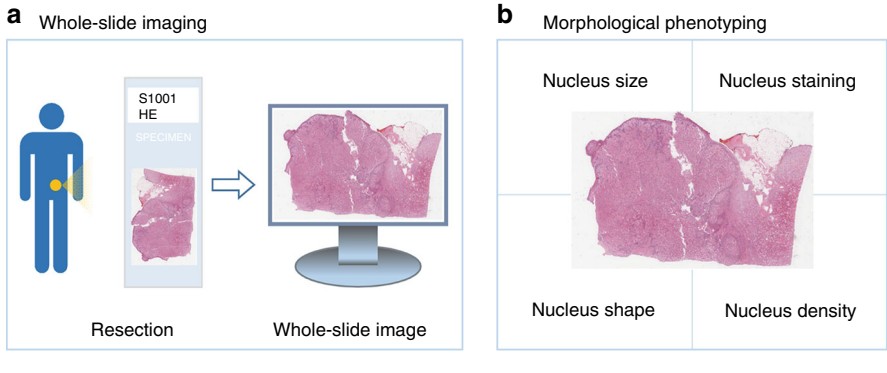

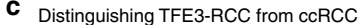

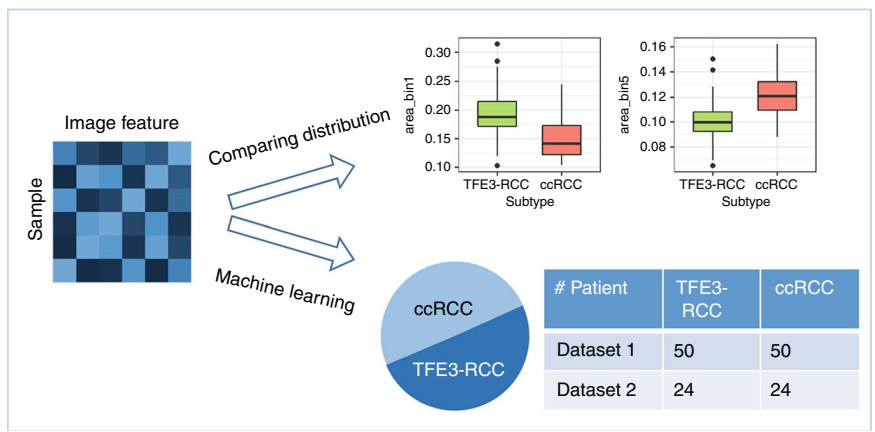

**Fig. 1 Workflow scheme. a** H&E-stained tissue slides were digitized by a scanner to obtain whole-slide images. **b** A large set of quantitative image features were extracted, characterizing nucleus size, staining, shape, and density. **c** The Mann–Whitney $U$ test was used to compare image features between TFE3-RCC and ccRCC, and machine learning models were developed based on the image features to automatically classify the two cancer subtypes. On the box plots in **c**, the central mark indicates the median, and the bottom and top edges of the box indicate the 25th and 75th percentiles ($q_1$ and $q_3$), respectively. The upper whisker extends from $q_3$ to $q_3 + 1.5 \times (q_3 - q_1)$, and the lower whisker extends from $q_1$ to $q_1 - 1.5 \times (q_3 - q_1)$, while data beyond the end of the whiskers are outlying points that are plotted individually.

each whole-slide image. More details can be found in the "Methods" section.

**Quantitative image features show significant differences between TFE3-RCC and ccRCC.** We applied Mann–Whitney $U$ test to each feature and identified 52 features significantly different between TFE3-RCC and ccRCC after multiple testing correction (5% false discovery rate; Fig. 3). Significant features were reported as overrepresented or underrepresented with respect to the TFE3-RCC subtype; i.e., a feature is defined as overrepresented if the median of this feature in TFE3-RCC group is higher than that in ccRCC group.

For the features related to nucleus size in Fig. 3, we found that area_bin1, area_bin9, and area_bin10 were overrepresented in TFE3-RCC whereas area_bin4, area_bin5, and area_bin6 were underrepresented. Image features from area_bin1 to area_bin10 represent the proportions of the nuclei with size varying from small to large. Therefore, these significant features indicate that the size of nucleus in TFE3-RCC is more heterogeneous and more towards the two extremes than that in ccRCC, which is also supported by the overrepresented feature, area_std (the standard deviation of nuclear size).

The features with names beginning with major, minor, and ratio in Fig. 3 are derived from the ellipses fitted to the segmented nuclei. These features are associated with nucleus shape. In particular, the features from ratio_bin1 to ratio_bin10 directly describe the percentages of the nuclei whose shape changes from

round to elongated. As we can see in Fig. 3, ratio_bin1 was underrepresented. In contrast, ratio_bin3, ratio_bin4, ratio_bin5, and ratio_std were overrepresented. Together, these observations suggest that ccRCC tends to have more nuclei that are very round.

Eleven nucleus staining-related features that were calculated in red and green channels showed significant difference between TFE3-RCC and ccRCC. Of those features, rMean_bin8, rMean_-bin9, rMean_mean, rMean_skewness, and gMean_mean were overrepresented for TFE3-RCC cases. rMean_bin8 and rMean_-bin9 represent the proportions of the nuclei that had very large mean pixel value in the red channel. rMean_mean and gMean_mean denote the average of mean pixel values of all nuclei in the red and green channels, respectively. rMean_skew-ness is overrepresented, implying that the data distribution of mean pixel values of nuclei in the red channel in TFE3-RCC was more asymmetric than that in ccRCC.

Of the 15 significant nucleus density-related features, we found five features overrepresented: distMin_bin1, distMin_bin2, dis-tMean_bin1, distMean_bin2, and distMax_bin1. The overrepre-sentation of the five features suggests that compared with ccRCC, TFE3-RCC tends to present more nuclei that are very close to each other. In other words, the cells in TFE3-RCC are more clumped together.

**Classification models based on image features effectively distinguish TFE3-RCC from ccRCC.** We first trained and evaluated our classifiers with five-fold cross-validation on dataset 1

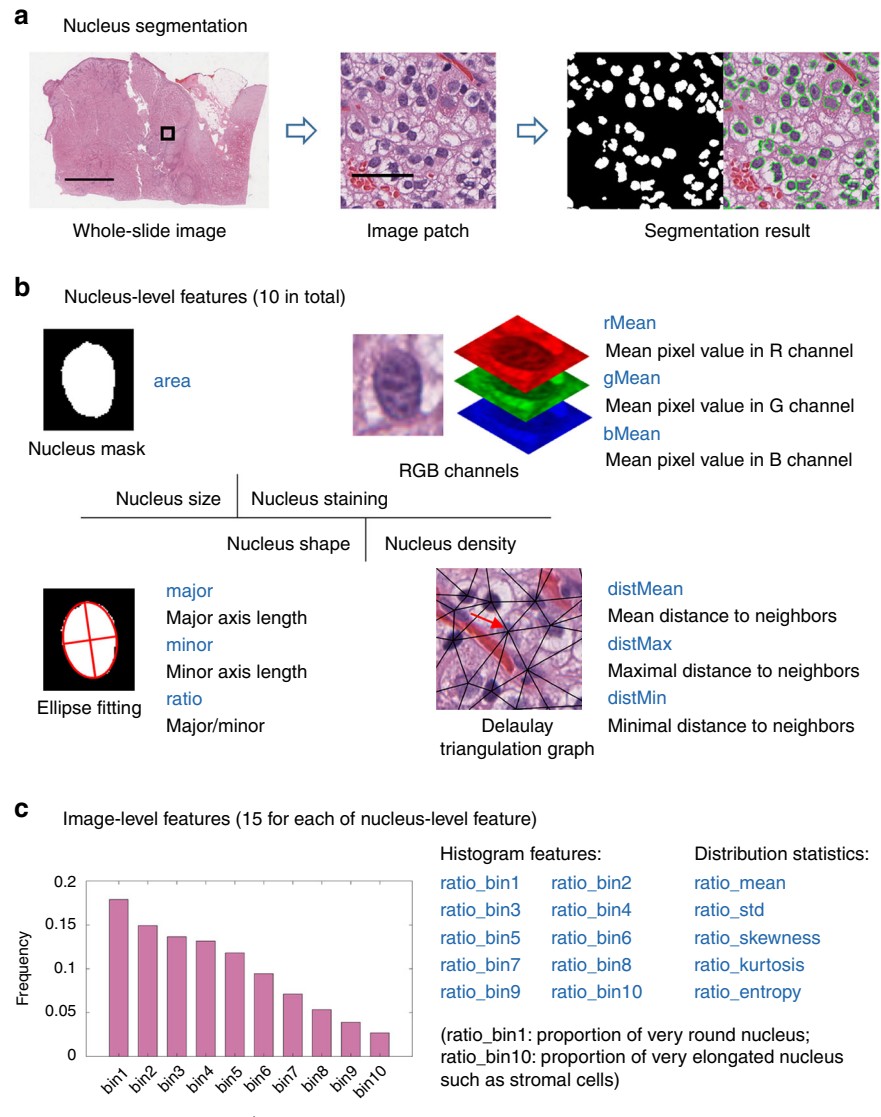

**Fig. 2 Feature extraction pipeline. a** The nuclei in whole-slide images are automatically segmented. **b** For each segmented nucleus, 10 nucleus-level features, regarding nucleus size, staining intensity, shape, and density, are extracted. **c** For each type of nucleus-level features from the same whole-slide image, they are dissected into 15 image-level features using a 10-bin histogram and five distribution statistics. Scale bars: 5 mm (**a** whole-slide image) and 50 μm (**a** image patch).

obtained from Indiana University (see Table 1 for details). In each of the five rounds, dataset 1 was randomly partitioned into two sets: 80% training and 20% internal validation. Our results showed that using the 30 features selected by the minimum redundancy maximum relevance (mRMR) algorithm, our best classifier, SVM with Gaussian kernel, attained an average AUC of 0.886. The performance of the four classifiers (logistic regression, SVM with linear kernel, SVM with Gaussian kernel, and random forest) did not differ significantly (ANOVA test $P$-value = 0.77). Bar graph of the results of five-fold cross-validation for four classifiers are shown in Fig. 4a.

The utility of our quantitative image features for diagnostic classification was further validated using an external dataset (dataset 2; Table 1). Specifically, we trained the same four classifiers using dataset 1 and then validated the performance using dataset 2. All classifiers achieved AUC that were similar to that obtained on the aforementioned internal cross-validation set (Fig. 4b). We also observed that, except for the random forest classifier, the other three classifiers achieved slightly higher AUC

on the external validation set than the average AUC of five-fold cross-validation using dataset 1. This may be because all patients in dataset 1 were used to train the classification models tested on dataset 2. In contrast, in five-fold cross-validation on dataset 1, only 80% of the patients in dataset 1 were used as the training set. The top quantitative features selected by mRMR (measured by feature importance score) included ratio_bin3, rMean_mean, minor_std, area_bin5, rMean_skewness, distMin_-bin5, rMean_std, and ratio_std.

## Discussion

To the best of our knowledge, this is the first study to provide a computational model to distinguish TFE3-RCC from ccRCC using quantitative histopathological features extracted from H&E-stained whole-slide images. In this study, we implemented an automated workflow that calculated 150 objective features from the images. The image features were extracted from the whole slides, which not only covered a large tumor area, but also

**Table 2 Illustrations of the 10 nucleus-level features.**

| Feature name | Interpretation | Patch with small value | Patch with large value |
|---|---|---|---|
| Area | Size of nucleus (unit: pixel) | 322 | 545 |
| Major | Major axis length (unit: pixel) | 26 | 30 |
| Minor | Minor axis length (unit: pixel) | 14 | 20 |
| Ratio | Major to minor ratio | 1.4 | 1.7 |
| rMean | Mean pixel value in R channel | 99 | 169 |
| gMean | Mean pixel value in G channel | 55 | 108 |
| bMean | Mean pixel value in B channel | 102 | 153 |
| distMean | Mean distance to neighbors (unit: pixel) | 60 | 95 |
| distMax | Maximal distance to neighbors (unit: pixel) | 86 | 123 |
| distMin | Minimal distance to neighbors (unit: pixel) | 31 | 41 |

The number beside each image patch is the mean feature value for all nuclei in the patch. For example, the number 322 means the mean area of the nuclei in the patch is 322 pixels. Scale bar: 50 μm for all patches.

covered a wide spectrum of cell nuclei morphology, including nucleus size, staining, shape, and density from the heterogeneous cancer tissue. We built and evaluated machine-learning models to classify patients into TFE3-RCC or ccRCC. The validity of this workflow is confirmed by an independent dataset collected from different sources.

Most cancers are heterogeneous and contain several subtypes[1,2]. Those subtypes are usually characterized by distinct molecular profiles that drive tumors to develop and progress differently[9–11]. Histopathology slides are routinely collected at the diagnosis of cancers. Our hypothesis is that tumor morphological phenotype can be detected quantitatively through artificial intelligence algorithms, which reflects underlying genetic aberrations including translocations. The TFE3-RCC is defined by the specific translocation on the cytoband Xp11.2. We reported, to the best of our knowledge, the largest TFE3-RCC cohort of 74 cases with an extensive analysis of the microscopic appearance of TFE3-RCC and ccRCC using computational pathological image analysis. Our results demonstrated the promising power of applying machine-learning models based on quantitative histopathological features to differentiate between TFE3-RCC and ccRCC, with impressive accuracy (AUC between 0.842 and 0.894) on the external

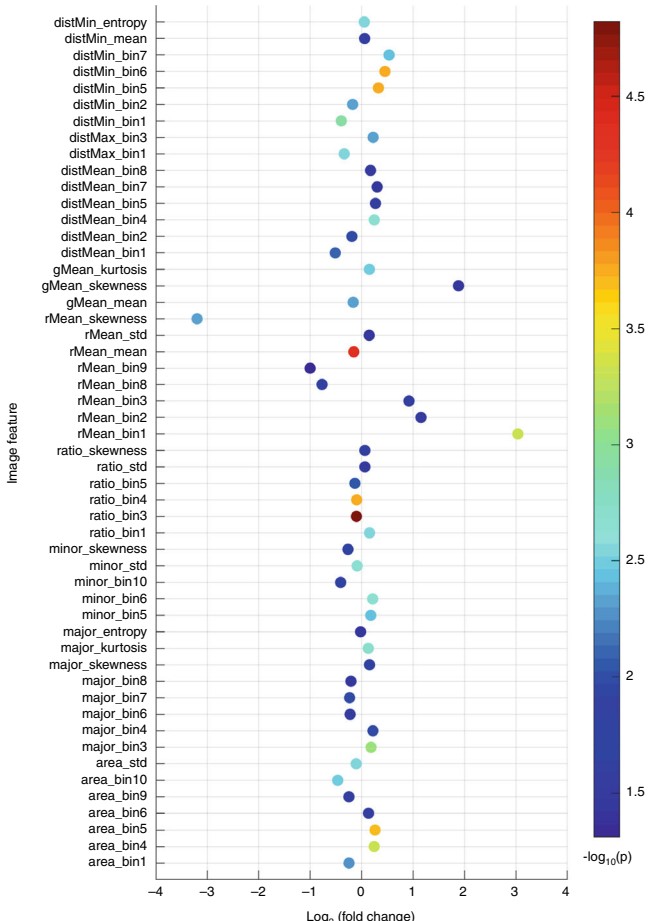

**Fig. 3 Comparison of image features between TFE3-RCC and ccRCC.** For each feature, the fold change is defined as the ratio of the median feature values between ccRCC and TFE3-RCC. 52 image features that show significant differences between TFE3-RCC and ccRCC are identified using the two-sided Mann–Whitney U test. Multiple comparison correction is performed using false discovery rate procedure at 5% level.

validation set. The strength of this tool will alleviate the underdiagnosis of TFE3-RCC and facilitate sample curation or clinical trial access directed at this group of patients.

We identified 52 image features significantly differing between the two subtypes. For example, in comparison with ccRCC, TFE3-RCC had higher proportions of very small and very large nuclei (see area_bin1, area_bin9, and area_bin10 in Fig. 3), which is in line with the fact that TFE3-RCC is more aggressive and associated with higher tumor grade[30] because high-grade tumors have faster cell proliferation rate. A senior pathologist (LC) was consulted on the significantly differing features. Although for some features it is difficult to tell their differences by human eyes, others can be visually perceived. For instance, we found that ccRCC had a higher proportion of very round nuclei (see ratio_bin1 in Fig. 3) than TFE3-RCC. The pathologist confirmed that ccRCC indeed tends to have rounder cell nuclei than TFE3-RCC. Another example was that the overrepresentation of our features (i.e., distMean_bin1 and distMean_bin2; Fig. 3) indicated more cell clumps in TFE3-RCC than ccRCC, which was also observed (Supplementary Fig. 1).

Since the TFE3 translocation causes overexpression of the TFE3 protein, immunohistochemistry (IHC) for TFE3 protein has been considered a surrogate for this genetic event. We compared the performance of our method with that from other reported studies using IHC. Sharain et al.[31] found in a two-laboratory study that the overall sensitivity and specificity of TFE3 IHC for TFE3-rearranged neoplasms was 85% and 57% at Laboratory A, and 70% and 95% at Laboratory B, leading to Youden indices of 0.42 and 0.65, respectively (Youden index = sensitivity + specificity−1). Their dataset contained 27 TFE3-rearranged neoplasms and 98 controls. Our SVM classifier with Gaussian kernel can achieve sensitivity of 91.7%, specificity of 79.2%, and Youden index of 0.708 (Fig. 4b). It is noteworthy that our pathological image-based classifier only relied on routine H&E staining instead of the staining of a specific molecule.

Previous studies investigating the clinicopathologic characteristics of TFE3-RCC often suffered from small sample size[32]. Our pathological image-based classifier can assist pathologists in diagnosing new TFE3-RCC cases and can also help in large-scale retrospective studies to retrieve old TFE3-RCC cases that were misdiagnosed. When used with an appropriate threshold, the classifier can automatically spot TFE3-RCC cases from the histopathology slide archive with very high sensitivity and relatively low false-positive rate (Fig. 4b). For instance, our SVM classifier with Gaussian kernel can achieve 91.7% sensitivity while retaining 20.8% false-positive rate. Given that the majority of RCC are ccRCC, its clinical application would allow pathologists to exclude many true negatives (ccRCC) for further evaluation or would nominate suspicious cases for further evaluation.

We also tested whether the differences in staining of H&E slides between institutions (thus different scanning instruments or slide preparation) would affect the generalization performance of our method. The slides in our external validation set (dataset 2) were from several institutions (University of Michigan and TCGA; TCGA cases themselves were also gathered from different institutions), and they had varied and different color appearances than the slides in dataset 1. We applied the same analysis workflow without the color normalization step and observed a large drop in generalization performance on the external validation set (Supplementary Fig. 2). This indicates that color normalization is a crucial step when dealing with whole-slide images from different sources.

In addition, we tested a convolutional neural network, ResNet-18, on dataset 1. The whole-slide images were resized to 224-by-224 pixels in order to feed into ResNet-18. The ResNet-18 was trained on 80% of all cases and validated on the remaining cases with five-fold cross-validation. Two training strategies were implemented, i.e., training the network from scratch and transfer learning. For transfer learning based on a pretrained ResNet-18 network, only the weights of the last two layers (the fully connected layer and softmax layer) were updated and the weights of earlier layers were kept frozen. The mean AUC generated from five-fold cross-validation is 0.518 for training from scratch and 0.696 for transfer learning. The performance of transfer learning is better, which may be due to far less parameters that need to be learned when using transfer learning. Compared with our classification models with AUCs between 0.8 and 0.9, ResNet-18's performance is inferior. It is well-known that the features learned by deep neural network are difficult to interpret. However, our classification pipeline is based on cellular image features, which are well-defined with clear meanings in cellular and tissue morphology and thus more interpretable and preferable in clinical diagnosis.

This study has several limitations. Intratumoral heterogeneity is a well-documented phenomenon in RCC[9–11]. Since we are unable to collect multiple formalin-fixed, paraffin-embedded tissue blocks from the same case, we cannot accurately evaluate intratumoral heterogeneity (ITH). Nonetheless, the whole-slide images were obtained from surgical resection specimens in our study. Surgical resection specimens cover a much larger area of a

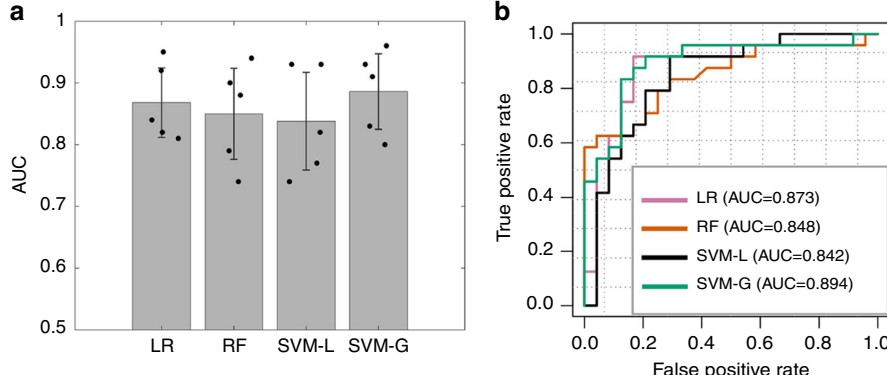

**Fig. 4 The performance of the four machine learning models. a** Classification performance on dataset 1 using five-fold cross-validation ($n = 5$ experiments for each model). For each, 80% of patients were used as the training set and the remaining patients were used as the internal validation set. **b** Receiver operating characteristics curves for classifying TFE3-RCC and ccRCC in the external validation set (dataset 2). Models were trained using dataset 1 and evaluated using dataset 2. The 95% confidence intervals for the AUC: LR (0.763–0.984), RF (0.736–0.960), SVM-L (0.725–0.959), and SVM-G (0.797–0.991). LR, logistic regression; RF, random forest; SVM-L, SVM with linear kernel; SVM-G, SVM with Gaussian kernel. Data are represented as mean ± SD in **a**.

tumor compared with needle biopsy. In addition, our algorithms take the ITH into consideration by using the distribution of the morphological characteristic values (histograms over ten bins) as imaging features. Although the consistently similar performance of the internal and external validation sets proves the stability and reproducibility of our imaging features and classification models, it would be more rigorous to demonstrate that these features are stable if evaluated from multiple sites of the same tumor. Another important limitation is that our study used matched ccRCC for comparison with TFE3-RCC. There are diverse morphologic manifestations of TFE3-RCC[1,2,5]. They also mimic papillary RCC, clear cell papillary RCC, unclassified RCC, chromophobe RCC, oncocytoma, and other rare renal tumors. Future studies should include other renal tumor types and histologic variants in matched cases for comparison.

In summary, we demonstrated that histopathology image classifiers based on quantitative features can successfully distinguish TFE3-RCC from ccRCC with a high accuracy (AUC of 0.894) on the external validation set, which corroborates our hypothesis that tumor histological phenotype can reflect underlying gene translocations. Our methods can facilitate TFE3-RCC diagnosis based on routinely collected H&E-stained histopathology slides, thereby contributing to accurate sample curation and treatment development of this rare and aggressive cancer subtype.

## Methods

**Sample collection**. Two datasets of H&E-stained whole-slide images (148 images in total) were collected. The ratio of TFE3-RCC patients to ccRCC patients was 1:1, and the gender and tumor grade information between the two subtypes were matched. Dataset 1 consisted of 50 TFE3-RCC patients and 50 ccRCC patients all from Indiana University. Dataset 2 was collected as an external validation set, containing 14 TFE3-RCC patients from the University of Michigan, 10 TFE3-RCC patients from TCGA[33], and 24 ccRCC patients from TCGA. All tumor samples were gathered by surgical excision. Tissue slides were scanned at ×40 magnification. No TFEB rearranged translocation RCC was included in the analysis. We did not attempt to subclassify TFE3-RCCs based on the rearrangement of TFE3 with different partner genes. Personal health information was de-identified in our datasets and hence this was an institutional review board approval–exempt study.

**Fluorescence in situ hybridization**. Interphase fluorescence in situ hybridization assay was performed on all tumors and described as follows[34–36]. The diagnosis of all TFE3-RCC cases were confirmed by FISH analysis. Specifically, tissue sections 4-μm thick were prepared from buffered formalin-fixed, paraffin-embedded tissue blocks containing tumor. The slides were deparaffinized with two washes with xylene (15 min each), and subsequently washed twice with absolute ethanol (10 min each), and then air dried in the hood. The slides were then treated with 10 mm citric acid (pH 6.0) (Zymed, San Francisco, CA, USA) at 95 °C for 10 min, rinsed in

distilled water for 3 min, and then washed with 2× SSC for 5 min. Digestion of the tissue was performed by applying 0.4 ml of pepsin (5 mg per ml in 0.01 N HCl and 0.9% NaCl) (Sigma, St Louis, MO, USA) at 37 °C for 40 min. The slides were rinsed with distilled water for 3 min, washed with 2× SSC for 5 min, and air dried. The split-apart probe set for TFE3 used BAC clones RP11-528A24 (116 kbp, located centromeric to TFE3, labeled with 5-fluorescein dUTP) and RP11-416B14 (182 kbp, located telomeric to TFE3, labeled with 5-ROX dUTP) (Empire Genomics, Buffalo, NY, USA). BAC clones for TFE3 were diluted with DenHyb2 at a ratio of 1:25. Diluted probe (5 μl) was applied to each slide in reduced light conditions. The slides were then covered with a 22 × 22-mm coverslip and sealed with rubber cement. Denaturation was achieved by incubating the slides at 83 °C for 12 min in a humidified box and hybridization at 37 °C overnight. The coverslips were removed, and the slides were washed twice with 0.1× SSC per 1.5 M urea at 45 °C (20 min each), and then washed with 2× SSC for 20 min and with 2× SSC per 0.1% NP-40 for 10 min at 45 °C. The slides were further washed with room temperature 2× SSC for 5 min. The slides were air dried and counterstained with 10 μl of 4′,6-diami-dino-2-phenylindole (Insitus), coverslipped, and sealed with nail polish.

The slides were examined with a Zeiss Axioplan 2 microscope (Zeiss, Göttingen, Germany). The images were acquired with a CMOS camera, and analyzed with metasystem software (MetaSystem, Belmont, MA, USA). Five sequential focus stacks with 0.4-mm intervals were acquired and then integrated into a single image to reduce thickness-related artifacts. For each case, a minimum of 100 tumor cell nuclei were examined with fluorescence microscopy at ×1000 magnification. Only non-overlapping tumor nuclei were evaluated. The TFE3 fusion resulted in a split-signal pattern. Signals were considered split when the green and red signals were separated by two or more signal diameters. On this basis and based on other commercially available break-apart FISH assays and TFE3 break-apart FISH assays, a positive result was reported when ≥10% of the tumor nuclei showed the split-signal pattern (Supplementary Fig. 3).

**Extraction of quantitative features from whole-slide images**. Each dimension of the whole-slide images ranged from about 40,000 to 130,000 pixels. The images were subdivided into tiles with the size of 2000 × 2000 to facilitate processing. Considering the color variations between institutions, before feature extraction we transformed the color appearance of the images in dataset 2 into that in dataset 1 using a structure-preserving color normalization algorithm[37]. To aggregate the nucleus-level features extracted from a patient into patient-level features, histograms and distribution statistics were employed. For constructing histogram features, a bag-of-visual-words model was utilized[38–40]. The bag-of-words model is a feature representation method originally used in natural language processing and information retrieval. In this model, a text is represented as a word-frequency histogram (i.e., each bin of the histogram represents the frequency of some word occurring in the text). This method has been widely adopted by computer vision in which image features are considered words. In this study, for each type of nucleus-level feature we create a histogram of the nucleus-level features. In this histogram, the words (i.e., midpoints of bins) are cluster centroids obtained by clustering nucleus-level features from the training set.

Specifically, for each type of nucleus-level feature, a large set of nucleus-level features were collected across patients from the training set and fed into K-means algorithm to learn 10 representative words (i.e., clustering centroids). The number of clusters is chosen using a cross-validation approach (Supplementary Fig. 4). After that, nucleus-level features extracted from a whole-slide image were assigned to their nearest bins using Euclidean distance, which resulted in a histogram of

word counts for each patient and for each type of nucleus-level features. The obtained histograms were L1-normalized to eliminate the impact of whole-slide images having different numbers of nuclei. As for distribution statistics, five parameters were calculated for each type of cell-level features; i.e., mean, standard deviation, skewness, kurtosis, and entropy. The entropy was computed based on the normalized histograms.

**Comparison of image feature distributions between TFE3-RCC and ccRCC.** To identify what specific image features showed distinct morphological differences between TFE3-RCC and ccRCC, we compared the distributions of each image feature between the two subtypes using a two-sided Mann–Whitney U test. To correct for multiple comparisons, we adjusted P values by the false discovery rate procedure according to Benjamini & Hochberg adjustment[41]. An adjusted P value < 0.05 was considered statistically significant.

**Machine-learning methods to classify TFE3-RCC and ccRCC.** Due to the high dimensionality of the image features and relatively small sample size, overfitting of the data is likely; therefore, before building classification models, we performed feature selection to avoid the overfitting problem. Feature dimensionality was reduced by the mRMR algorithm[42] using R package mRMRe. mRMR has been shown to be a robust feature selection algorithm in various tasks[43–45]. The mRMR algorithm was applied to all image features with regard to the class label of sample (i.e., TFE3-RCC or ccRCC) to select an informative and non-redundant set of features.

Logistic regression, SVM with linear or Gaussian kernels, and random forest were used to conduct supervised machine learning. R version 3.5 was used to train and test classification models, with glmnet package for logistic regression, randomForest package for random forest, and e1071 package for SVM. In dataset 1, five-fold cross-validation was used. To further validate our method using an external validation set, classification models were trained using dataset 1 and evaluated using dataset 2. AUC and confidence intervals were computed with the R package pROC.

## Data availability
The quantitative image features extracted from H&E stained whole-slide images are available from GitHub at (https://github.com/chengjun583/tRCC-ccRCC-classification). The remaining data is available in the Article, Supplementary Information files or available from the authors upon reasonable request.

## Code availability
The source code of this work can be downloaded from GitHub at (https://github.com/chengjun583/tRCC-ccRCC-classification).

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

## Acknowledgements

This work was supported in part by American Cancer Society Institutional Research Grant to Indiana University (J.Z.), National Natural Science Foundation of China (No. 61901275), National Key R&D Program of China (No. 2019YFC0118300), Shenzhen Peacock Plan (KQTD2016053112051497 and KQJSCX20180328095606003), Indiana University Precision Health Initiative, Young Faculty Support Program of SZU Health Science Center (No. 71201-000001), Natural Science Foundation of SZU (No. 2019131), and Medical Scientific Research Foundation of Guangdong Province, China (No. B2018031).

## Author contributions

J.C., L.C., K.H., and J.Z. conceived and designed the study. J.C. and Z.H. performed the computational analysis with assistance from W.S., R.M., M.C., Q.F., and D.N. The paper was written by J.C., J.Z., and K.H. with contributions by all co-authors.

## Competing interests

The authors declare no competing interests.
