## [Peer Review File · Nature Communications]

Reviewers' comments:

Reviewer #1 (Remarks to the Author): Expert in kidney cancer treatment

The authors present a novel method for predicting the presence of Xp11.2 translocation renal cell carcinoma (tRCC) by a image analysis, utilising machine learning approach utilising archived diagnostic H&E stained histopathology sections.

Methods: the authors obtain H&E sections from 50 Xp11.2 tRCC patients and confirm the diagnosis utilising FISH (data not shown). 60 H&E sections from ccRCC patients serve as controls. A pathological image analysis pipeline extracted quantitative image features from whole slide images characterising the size, staining intensity, shape and density of cell nuclei (10 nucleus level features extracted) and further for each segmented nucleus-level feature, 15 image-level features are extracted from the whole-slide image. 150 image features are taken forward for feature correlation/selection using the Mann-Whitney U test to compare distinguishing features between tRCC and ccRCC. A lasso logistic regression model is trained based on the image features to automatically classify the two cancer types in a training and validation cohort.

The study addresses a shortcoming in the current pathway for histopathological diagnosis of tRCC (a rare clinical entity with poor prognosis) which is likely underdiagnosed.

- An image analysis tool that could screen cases for further review and validation testing by a pathologist could be clinically useful.
- We note that this cohort represents the largest reported cohort of tRCC in the literature.
- The proposed algorithm is unique and solid. The combination of MRMR and lasso logistic regression for filtering features of tRCC against ccRCC is a practical and innovative approach from a machine learning perspective.
- Nuclear characterisation means that this analysis is 'simple' and robust and therefore likely to be relatively easy to implement clinically.

As such we deem the study to be novel, timely and clinically pertinent. We would recommend that the following issues are resolved/addressed in a revised manuscript prior to recommending publication.

- Independent validation cohort: although the cohort is split into training and validation, the study would benefit from an independent validation cohort. This is a single institution study and differences in staining/preparation of H&E slides between institutions may make the algorithm less generalisable to other centres. Did the authors consider accessing a publicly available dataset such as the TCGA?
- Nuclei heterogeneity: heterogeneity in nuclear size is a key feature in the analysis. Given the well documented phenomenon of intratumoural heterogeneity (ITH), it would be important to demonstrate that these features are pervasive/stable if evaluated from multiple sites in the same patient to demonstrate there is no bias. If there is a bias, the technique is not useful. The authors should consider multiple slides/regions from each case to test the stability of their model. This weakness in the study design must be addressed prior to publication.
- Diagnosis of tRCC: the authors describe FISH being performed to confirm the diagnosis- these data establishing the ground truth are not presented and the authors should present their methodology and the data to support this.
- Discussion: the authors need to give greater prominence to the fact that the tool has utility in nominating cases for further evaluation by the pathologist
- Style: the manuscript would benefit generally from a native speaker edit

Minor comments:

Line 55: Ref 13. Should read Choueri et al

Line 55-58: Currently there are no 'therapeutic implications' of a tRCC diagnosis and this sentence should be revised. The referenced retrospective observational study demonstrated that although

VEGF targeted therapy demonstrated some clinical activity in advanced tRCCs, the response rate and survival of the cohort overall was poor, highlighting the poor prognosis associated with this rare subtype of RCC. The strength of this tool would be in improving underdiagnosis and facilitating data collection or clinical trial access directed at this group of patients for example.

Reviewer #2 (Remarks to the Author): Expert in kidney cancer pathology

In this study the authors have used whole slide images and machine learning to develop an algorithm to distinguish Xp11.2 translocation associated renal cell carcinoma (tRCC) from clear cell renal cell carcinoma (cRCC).

A major concern is the very different grade distribution between both cohorts. 34 of 42 (81%) "gradeable" tRCC were high grade (3 or 4) while only 27 of 57 (47%) of "gradeable" cRCC were of similar grade. Given this selection bias, it is predictable that nuclear size is different in both groups. At the very least, the cRCC cohort should include a grade distribution similar to the tRCC cohort. Grade can also affect nuclear shape and nuclear chromatin irregularities (polychromasia) so any differences found between the cohorts would be expected. This also applies to the possibility of nuclear overlap since this finding would also be expected with an increase in the nuclear to cytoplasmic ratio, as seen in higher grade tumors. There is no reason why grade should be unavailable for any case.

It is surprising that stromal characteristics were not used in developing the model. For the most part clear cRCC, particularly those of lower grade, have a relative unique vascular network which is rarely encountered in non-cRCC.

I agree with the authors that a limitation of the study is the fact that tRCC were only compared to cRCC, even though some of the former enter in the differential diagnosis of renal carcinomas with papillary growth and/or cytoplasmic eosinophilia.

Reviewer #3 (Remarks to the Author): Expert in machine learning

This study focuses on Renal Cell Carcinoma (RCC) and more specifically on the tRCC subtype which is characterized by Xp11.2 translocations. Patients with this translocation have been found to respond to targeted therapies and therefore identifying this subtype from the other RCC subtypes is important. Identifying this subtype from pathological images is challenging due to its similarities with other types of RCC and there is high risk of misclassification with subsequent delays in therapy. To address this, the authors are using machine learning on pathological images to identify tRCC unique image features to distinguish between tRCC and ccRCC and build a classification model based on the identified features.

The authors assembled their own cohort of 50 Xp11.2 tRCC patients and 60 ccRCC patients with one image per patient and scanned them at 40x magnification. They extracted quantitative image features from the whole image. These features were the size, staining, shape and density of cell nuclei. To compare the feature distributions between tRCC and ccRCC, they used the Mann-Whitney U test and then they built a lasso logistic regression model for patient classification into tRCC or ccRCC.

This work is of potential interest, but I have several concerns, described below:

In Table 1, it appears that the tRCC and ccRCC cohorts have significant differences in demographic

and tumor characteristics, raising the possibility that the proposed model may (partially) recapitulate these differences. The authors need to correct for these biases.

In Table 2, it might be more informative if there were schematics on the image patches demonstrating the different values of the features, such as lines for axis or circles for the areas to visually understand what is the difference between a "small" and a "large" value. Or the actual feature values for each patch.

Fig 2C and the "Extraction of quantitative features from whole-slide images" paragraph in the Methods section are a little difficult to understand. Maybe explain a little better what are the "words" or "cluster centers" in this scenario. Also, the concept of a "bin" seems to be very important in the paper after this part so it will be beneficial if there was a more thorough explanation of the binning process. How robust are the results to the binning procedure?

During feature extraction, it appears that all patients have been used: does this include the independent test set? To prevent data leakage, feature extraction should be performed only on the training set.

Figure 3: replace it with a plot that shows fold enrichment and p value simultaneously for each feature, e.g.:
https://yulab-smu.github.io/clusterProfiler-book/clusterProfiler_files/figure-html/unnamed-chunk-48-1.png

It would be important to ask a pathologist to review the image patches to verify that the features that are identified as important by the model are indeed observed in the images.

The data we split equally and randomly in training and testing. Utilizing cross-validation could help establish the performance of the classifier in different data splits and strengthen the argument.

The authors could try to use other machine learning models such as decision trees or SVMs and compare the performance to logistic regression. Also, although the dataset is small, I suggest that the authors try a simple convolutional neural network, e.g. Resnet-18. If successful, it would be very interesting to compare the per tile Resnet-18 vs elastic net scores (across all tiles) to test whether Resnet and elastic net detect similar features.

The most important limitation of this study is the lack of an external validation cohort and the comparison of tRCC only to ccRCC and not other types of RCC. The authors need to find a way to overcome this limitation. I propose utilizing TCGA. Based on what is mentioned by the authors, I understand that TCGA may have very few tRCC cases. However, the authors can still apply their model on all the TCGA cases and check whether the translocated cases rank high compared to others. Given the high AUC in their cohort, it would be expected that the results will not be too bad.

Also, please calculate confidence intervals for AUC and other reported metrics.

Finally, the introduction needs to be updated with more recent results:

<https://www.ncbi.nlm.nih.gov/pubmed/30224757>

<https://www.ncbi.nlm.nih.gov/pubmed/31160815>

<https://www.ncbi.nlm.nih.gov/pubmed/31308507>

and many others.

Response to Reviewers

Reviewer #1 (Remarks to the Author): Expert in kidney cancer treatment

The authors present a novel method for predicting the presence of Xp11.2 translocation renal cell carcinoma (tRCC) by a image analysis, utilising machine learning approach utilising archived diagnostic H&E stained histopathology sections.

Methods: the authors obtain H&E sections from 50 Xp11.2 tRCC patients and confirm the diagnosis utilising FISH (data not shown). 60 H&E sections from ccRCC patients serve as controls. A pathological image analysis pipeline extracted quantitative image features from whole slide

images characterising the size, staining intensity, shape and density of cell nuclei (10 nucleus level features extracted) and further for each segmented nucleus-level feature, 15 image-level features are extracted from the whole-slide image. 150 image features are taken forward for feature correlation/selection using the Mann-Whitney U test to compare distinguishing features between tRCC and ccRCC. A lasso logistic regression model is trained based on the image features to automatically classify the two cancer types in a training and validation cohort.

The study addresses a shortcoming in the current pathway for histopathological diagnosis of tRCC (a rare clinical entity with poor prognosis) which is likely underdiagnosed.

- An image analysis tool that could screen cases for further review and validation testing by a pathologist could be clinically useful.

- We note that this cohort represents the largest reported cohort of tRCC in the literature.

- The proposed algorithm is unique and solid. The combination of MRMR and lasso logistic regression for filtering features of tRCC against ccRCC is a practical and innovative approach from a machine learning perspective.

- Nuclear characterisation means that this analysis is ‘simple’ and robust and therefore likely to be relatively easy to implement clinically.

As such we deem the study to be novel, timely and clinically pertinent. We would recommend that the following issues are resolved/addressed in a revised manuscript prior to recommending publication.

Response: We thank the reviewer for the recognition of the impact and contribution of our manuscript.

Comment: Independent validation cohort: although the cohort is split into training and validation, the study would benefit from an independent validation cohort. This is a single institution study and differences in staining/preparation of H&E slides between institutions may make the algorithm less generalisable to other centres. Did the authors consider accessing a publicly available dataset such as the TCGA?

Response: We thank the reviewer for this valuable suggestion. Testing on an independent validation cohort would make our results and conclusions more solid, so we collected additional cases from the University of Michigan and TCGA as an independent validation set. The new cases include 24 tRCC and 24 ccRCC. We demonstrated that our classification models generalize well on the independent validation set, with AUC between 0.842 and 0.894 (see Fig. 4b).

Comment: Nuclei heterogeneity: heterogeneity in nuclear size is a key feature in the analysis. Given the well documented phenomenon of intratumoural heterogeneity (ITH), it would be important to demonstrate that these features are pervasive/stable if evaluated from multiple sites in the same patient to demonstrate there is no bias. If there is a bias, the technique is not useful. The authors should consider multiple slides/regions from each case to test the stability of their model. This weakness in the study design must be addressed prior to publication.

Response: We agree that intratumoural heterogeneity is a common issue in cancer-related studies.

In our study, the whole-slide images were obtained from surgical resection specimens. Whole slide images cover a much larger area of a tumor. We extracted image features based on all cell nuclei in a whole-slide image instead of, like some other studies did, extracting features from a few selected image patches. In addition, our algorithms take the ITH into consideration by using the distribution of the morphological characteristic values (histograms over ten bins) as imaging features. By doing so, we believe the intratumoral heterogeneity issue can be greatly alleviated. Also, the consistently similar performance on both internal validation and external validation sets proves the stability and reproducibility of our model and workflow.

Comment: Diagnosis of tRCC: the authors describe FISH being performed to confirm the diagnosis- these data establishing the ground truth are not presented and the authors should present their methodology and the data to support this.

Response: *We added a subsection titled “Fluorescence in situ hybridization” in the Methods section, which provides details about the diagnosis of Xp11.2 tRCC using break-apart FISH assay. All the tRCC study cases were confirmed by FISH analysis.*

Comment: Discussion: the authors need to give greater prominence to the fact that the tool has utility in nominating cases for further evaluation by the pathologist.

Response: *We have revised the Discussion section to emphasize the utility of our tool in nominating cases for further evaluation (lines 228-236).*

Comment: Style: the manuscript would benefit generally from a native speaker edit.

Response: *We had a native speaker edit the manuscript.*

Minor comments:

Comment: Line 55: Ref 13. Should read Choueri et al

Response: *We thank the reviewer for pointing out this mistake. This has been corrected in the revised manuscript.*

Comment: Line 55-58: Currently there are no ‘therapeutic implications’ of a tRCC diagnosis and this sentence should be revised. The referenced retrospective observational study demonstrated that although VEGF targeted therapy demonstrated some clinical activity in advanced tRCCs, the response rate and survival of the cohort overall was poor, highlighting the poor prognosis associated with this rare subtype of RCC. The strength of this tool would be in improving underdiagnosis and facilitating data collection or clinical trial access directed at this group of patients for example.

Response: *This sentence has been revised, and changes are highlighted in yellow in the revised manuscript (lines 61-65).*

Reviewer #2 (Remarks to the Author): Expert in kidney cancer pathology

In this study the authors have used whole slide images and machine learning to develop an algorithm to distinguish Xp11.2 translocation associated renal cell carcinoma (tRCC) from clear cell renal cell carcinoma (cRCC).

Comment: A major concern is the very different grade distribution between both cohorts. 34 of 42 (81%) “gradeable” tRCC were high grade (3 or 4) while only 27 of 57 (47%) of “gradeable” cRCC were of similar grade. Given this selection bias, it is predictable that nuclear size is different in both groups. At the very least, the cRCC cohort should include a grade distribution similar to the tRCC cohort. Grade can also affect nuclear shape and nuclear chromatin irregularities (polychromasia) so any differences found between the cohorts would be expected. This also applies to the possibility of nuclear overlap since this finding would also be expected with an increase in the nuclear to cytoplasmic ratio, as seen in higher grade tumors. There is no reason why grade should be unavailable for any case.

Response: *Previously a few cases did not have tumor grade information. In the revised version, we had a pathologist (Prof. Liang Cheng) grade those cases to complete tumor grade information. We collected additional new cases from Indiana University and combined with the previously collected cases so that the newly combined local cohort contains 50 tRCC cases and 50 ccRCC cases with matched patient characteristics (gender and tumor grade). This cohort constitutes the training set and the internal validation (cross-validation) set. In addition, we collected an external validation set from the University of Michigan and TCGA, including 24 tRCC and 24 ccRCC, which have matched gender and tumor grade. The results showed that 52 image features significantly differed between the tRCC and ccRCC cohorts (see Fig. 3). After correction for the imbalance of tumor grade, our classification models can still effectively differentiate between tRCC and ccRCC (see Fig. 4).*

Comment: It is surprising that stromal characteristics were not used in developing the model. For the most part clear cRCC, particularly those of lower grade, have a relative unique vascular network which is rarely encountered in non-cRCC.

Response: *Stromal cells such as fibroblasts are typically spindle-shaped with elongated nuclei and therefore characterized by long major axis and/or large ratio between major axis and minor axis, so among the extracted 150 image features, some features, such as ratio_bin10 and ratio_bin9 (large ratio), are directly related to tumor stroma. In the results of Mann-Whitney U test for each feature, we observed that some stroma-related features, such as ratio_bin10 and ratio_bin, showed significant difference between tRCC and ccRCC (unadjusted P value = 0.034 and 0.0182, respectively). However, these features are not significant enough to be selected by the feature selection algorithm to develop the classification model.*

Comment: I agree with the authors that a limitation of the study is the fact that tRCC were only compared to cRCC, even though some of the former enter in the differential diagnosis of renal

carcinomas with papillary growth and/or cytoplasmic eosinophilia.

Response: We thank the reviewer for pointing out this fact. Considering the fact that the majority of RCC are ccRCC, we currently only used ccRCC as the control group and would include other less common subtypes into comparison group in our future work.

Reviewer #3 (Remarks to the Author): Expert in machine learning

This study focuses on Renal Cell Carcinoma (RCC) and more specifically on the tRCC subtype which is characterized by Xp11.2 translocations. Patients with this translocation have been found to respond to targeted therapies and therefore identifying this subtype from the other RCC subtypes is important. Identifying this subtype from pathological images is challenging due to its similarities with other types of RCC and there is high risk of misclassification with subsequent delays in therapy. To address this, the authors are using machine learning on pathological images to identify tRCC unique image features to distinguish between tRCC and ccRCC and build a classification model based on the identified features.

The authors assembled their own cohort of 50 Xp11.2 tRCC patients and 60 ccRCC patients with one image per patient and scanned them at 40x magnification. They extracted quantitative image features from the whole image. These features were the size, staining, shape and density of cell nuclei. To compare the feature distributions between tRCC and ccRCC, they used the Mann-Whitney U test and then they built a lasso logistic regression model for patient classification into tRCC or ccRCC.

This work is of potential interest, but I have several concerns, described below:

Comment: In Table 1, it appears that the tRCC and ccRCC cohorts have significant differences in demographic and tumor characteristics, raising the possibility that the proposed model may (partially) recapitulate these differences. The authors need to correct for these biases.

Response: To avoid possible biases, we collected new cases to make the tRCC and ccRCC cohorts have matched gender and tumor grade and repeated the analysis. We found that our classification models could still effectively distinguish tRCC from ccRCC. For example, the SVM classifier with Gaussian kernel achieved a mean AUC of 0.886 using five-fold cross validation.

Comment: In Table 2, it might be more informative if there were schematics on the image patches demonstrating the different values of the features, such as lines for axis or circles for the areas to visually understand what is the difference between a “small” and a “large” value. Or the actual feature values for each patch.

Response: To make Table 2 more informative, we added the actual feature value for each patch in the revised manuscript. In addition, Figure 2B provides an intuitive explanation of what these 10 nucleus-level features mean.

Comment: Fig 2C and the “Extraction of quantitative features from whole-slide images”

paragraph in the Methods section are a little difficult to understand. Maybe explain a little better what are the “words” or “cluster centers” in this scenario. Also, the concept of a “bin” seems to be very important in the paper after this part so it will be beneficial if there was a more thorough explanation of the binning process. How robust are the results to the binning procedure?

Response: *The bag-of-words model is a feature representation method originally used in natural language processing and information retrieval. In this model, a text is represented as a word-frequency histogram (i.e., each bin of the histogram represents the frequency of some word occurring in the text). This method was later adopted in computer vision and image analysis where image features are treated as “words”. In this study, for each type of nucleus-level feature we create a histogram of the nucleus-level features. In this histogram, the “words” (i.e., midpoints of bins) are cluster centroids obtained by clustering nucleus-level features from the training set.*

In essence, the bag-of-words model in this scenario is like the process of creating a conventional histogram for numeric data (grouping the data into bins). However, unlike conventional histograms where the midpoints of bins are uniformly distributed, in the bag-of-words model the midpoints of bins are cluster centroids (or modes) obtained by clustering the numeric data (nucleus-level features in our case) using clustering algorithms. Compared with conventional histograms, the advantage of the bag-of-words model is that it has smaller quantization error (creating a histogram can be considered as quantization—replacing values within the range of each bin with its mid-point value). Empirically, the results are robust when the number of bins is set to around 10. Too few or too many bins will deteriorate the performance. We have revised the Methods section to make the feature extraction part more clear.

Comment: During feature extraction, it appears that all patients have been used: does this include the independent test set? To prevent data leakage, feature extraction should be performed only on the training set.

Response: *We thank the reviewer for pointing out this confusion. We only used the patients from the training set during feature extraction and model training, and left the test set untouched. We have revised the text to avoid potential misunderstanding.*

Comment: Figure 3: replace it with a plot that shows fold enrichment and p value simultaneously for each feature, e.g.:

https://yulab-smu.github.io/clusterProfiler-book/clusterProfiler_files/figure-html/unnamed-chunk-48-1.png

Response: *We thank the reviewer for this great suggestion. We have replaced it with a more intuitive and informative plot that shows fold change and P value simultaneously (see Fig. 3).*

Comment: It would be important to ask a pathologist to review the image patches to verify that the features that are identified as important by the model are indeed observed in the images.

Response: *Prof. Liang Cheng, an expert genitourinary pathologist and one of the co-authors in this study, has reviewed the image patches to verify the observed features in current study. For*

example, ccRCC tends to have rounder nuclei than tRCC (see the feature ratio_bin1 in Figure 3). Another example is that more cell clumps are observed in tRCC as indicated by the features distMean_bin1 and distMean_bin2 (Fig. 3; also see the image patches in Supplementary Fig. 1).

Comment: The data we split equally and randomly in training and testing. Utilizing cross-validation could help establish the performance of the classifier in different data splits and strengthen the argument.

Response: *We thank the reviewer for his suggestion. To avoid possible bias of one-time data split, we have used five-fold cross validation to evaluate the performance of the classifiers. The mean AUC of the classification models with the five-fold cross validation ranged from 0.838 to 0.886 (see Fig. 4a).*

Comment: The authors could try to use other machine learning models such as decision trees or SVMs and compare the performance to logistic regression. Also, although the dataset is small, I suggest that the authors try a simple convolutional neural network, e.g. Resnet-18. If successful, it would be very interesting to compare the per tile Resnet-18 vs elastic net scores (across all tiles) to test whether Resnet and elastic net detect similar features.

Response: *In addition to logistic regression we already used, we tested three other machine learning models: SVM with Gaussian kernel, SVM with linear kernel, and random forest. These models achieved AUC between 0.842 and 0.894 on an external validation set (Fig. 4b). The Resnet-18 network was tested but its training was not successful with much worse performance than traditional machine learning models due to the small sample size. The performance comparison is included in the Results section (line 164) in the manuscript and highlighted in yellow.*

Comment: The most important limitation of this study is the lack of an external validation cohort and the comparison of tRCC only to ccRCC and not other types of RCC. The authors need to find a way to overcome this limitation. I propose utilizing TCGA. Based on what is mentioned by the authors, I understand that TCGA may have very few tRCC cases. However, the authors can still apply their model on all the TCGA cases and check whether the translocated cases rank high compared to others. Given the high AUC in their cohort, it would be expected that the results will not be too bad.

Response: *we collected an external validation set from the University of Michigan and TCGA, including 24 tRCC and 24 ccRCC, which have matched gender and tumor grade. Analysis results showed that our classification models generalize well on the external validation set, achieving AUC between 0.842 and 0.894 (see Fig. 4b). Since the majority of RCC are ccRCC, we currently only used ccRCC as control group and would include other less common subtypes for comparison in our future work.*

Comment: Also, please calculate confidence intervals for AUC and other reported metrics.

Response: *We thank the reviewer for this suggestion and added 95% confidence intervals for*

reported metrics (see Fig. 4b).

Comment: Finally, the introduction needs to be updated with more recent results:

<https://www.ncbi.nlm.nih.gov/pubmed/30224757>

<https://www.ncbi.nlm.nih.gov/pubmed/31160815>

<https://www.ncbi.nlm.nih.gov/pubmed/31308507>

and many others.

Response: *We thank the reviewer for pointing out the recent results. We have added these articles in the Introduction and References section.*

<https://www.ncbi.nlm.nih.gov/pubmed/30224757> now ref No.28

<https://www.ncbi.nlm.nih.gov/pubmed/31160815> now ref No.27

<https://www.ncbi.nlm.nih.gov/pubmed/31308507> now ref No.23

REVIEWERS' COMMENTS:

Reviewer #1 (Remarks to the Author):

The authors have gone some way to address the concerns , namely there is now an independent validation cohort, demonstrating reproducibility of their model on samples from a separate institution and a TCGA cohort. Some issues remain:

The ITH issue was not resolved - using WSI is not akin to assessing multiple spatially separate areas of the tumour. If the authors do not have access to multiple FFPE blocks from the same cases that is understandable but they should be explicit in their inability to accurately evaluate ITH.

Regarding this statement: "The Resnet-18 network was tested but its training was not successful with much worse performance than traditional machine learning models due to the small sample size."

It is unclear if the author mean that they could not obtain a training model or get reliable results? Resnet 18 is workable for small sample size, so they should clarify exactly the outcome of the testing was.

Reviewer #2 (Remarks to the Author):

The authors have revised the manuscript significantly and have addressed most of my concerns. There are a few small issues which need to be corrected.

The authors use the term tRCC throughout the manuscript. They should change this term for TFE3 tRCC since the family of translocated renal tumors include others, such as those with TFEB translocations. The authors should also state that they made no attempt to subclassify their TFE3 translocated tumors since Xp11 can be rearranged to multiple partners. The authors should specifically state that no TFEB rearranged renal tumors were included in the analysis.

Once again, I state that it is a pity that the analysis only included clear cell RCC as a comparator. Although they are the largest group, many TFE3 tRCC have non-clear cytoplasm.

Reviewer #3 (Remarks to the Author):

The authors have done a good job revising the manuscript.

I have one remaining point, regarding the robustness of the number of bins. In their rebuttal, the authors say: "Empirically, the results are robust when the number of bins is set to around 10":

- Is this number chosen using a cross-validation approach? It needs to be done this way.
- Can you show a plot of AUC vs number of bins?

Re: NCOMMS-19-29922B “Computational analysis of pathological images enables a better diagnosis of TFE3 Xp11.2 translocation renal cell carcinoma”

Reviewer #1 (Remarks to the Author):

The authors have gone some way to address the concerns, namely there is now an independent validation cohort, demonstrating reproducibility of their model on samples from a separate institution and a TCGA cohort. Some issues remain:

Comment 1: The ITH issue was not resolved - using WSI is not akin to assessing multiple spatially separate areas of the tumour. If the authors do not have access to multiple FFPE blocks from the same cases that is understandable but they should be explicit in their inability to accurately evaluate ITH.

Response: We appreciate the reviewer’s understanding and suggestion. Since we are unable to collect multiple FFPE blocks from the same cases to evaluate ITH accurately, we have stated this limitation in the Discussion section.

“This study has several limitations. Intratumoral heterogeneity is a well-documented phenomenon in RCC⁹⁻¹¹. Since we are unable to collect multiple formalin-fixed, paraffin-embedded tissue blocks from the same case, we cannot accurately evaluate intratumoral heterogeneity (ITH). Nonetheless, the whole-slide images were obtained from surgical resection specimens in our study. Surgical resection specimens cover a much larger area of a tumor compared with needle biopsy. In addition, our algorithms take the ITH into consideration by using the distribution of the morphological characteristic values (histograms over ten bins) as imaging features. Although the consistently similar performance of the internal and external validation sets proves

the stability and reproducibility of our imaging features and classification models, it would be more rigorous to demonstrate that these features are stable if evaluated from multiple sites of the same tumor.” (lines 269-279)

Comment 2: Regarding this statement: “The Resnet-18 network was tested but its training was not successful with much worse performance than traditional machine learning models due to the small sample size.” It is unclear if the author mean that they could not obtain a training model or get reliable results? Rsenet 18 is workable for small sample size, so they should clarify exactly the outcome of the testing was.

Response: *We applied Resnet18 model on our data analysis. It shows a much lower AUC performance than our method. Resnet18 was implemented using MATLAB 2019b Deep Learning Toolbox and evaluated on dataset 1 via the following procedure. Since the resnet18 has an input size of 224-by-224, the whole-slide images in dataset 1 were resized to 224-by-224 pixels. The resnet18 was trained on 80% of all cases and validated on the remaining cases with five-fold cross validation. Two training strategies were implemented, i.e., training the network from scratch and transfer learning. For transfer learning based on a pretrained resnet18 network, only the weights of the last two layers (the fully connected layer and softmax layer) were updated and the weights of earlier layers were kept frozen. The mean AUC generated from five-fold cross validation is 0.518 for training from scratch and 0.696 for transfer learning. The performance of transfer learning is better, which may be due to far less parameters that need to be learned when using transfer learning. However, compared to our classification models with AUCs close to 0.9, Resnet 18’s performance is inferior. This test on Resnet 18 is discussed in the Discussion section (lines 255-268).*

Reviewer #2 (Remarks to the Author):

The authors have revised the manuscript significantly and have addressed most of my concerns. There are a few small issues which need to be corrected.

Comment 3: The authors use the term tRCC throughout the manuscript. They should change this term for TFE3 tRCC since the family of translocated renal tumors include others, such as those with TFEB translocations. The authors should also state that they made no attempt to subclassify their TFE3 translocated tumors since Xp11 can be rearranged to multiple partners. The authors should specifically state that no TFEB rearranged renal tumors were included in the analysis.

Response: We follow the reviewer's suggestion and have changed "tRCC" to "TFE3-RCC" throughout the manuscript. We also revised our Method section to clarify the issues raised by the reviewer.

"No TFEB rearranged translocation RCC was included in the analysis. We did not attempt to subclassify TFE3-RCCs based on rearrangement of TFE3 with different partner genes." (lines 300-302)

Comment 4: Once again, I state that it is a pity that the analysis only included clear cell RCC as a comparator. Although they are the largest group, many TFE3 tRCC have non-clear cytoplasm.

Response: We address the reviewer's concern in our revised manuscript.

“Another important limitation is that our study used matched ccRCC for comparison with TFE3-RCC. There are diverse morphologic manifestations of TFE3-RCC^{1,2, 5}. They also mimic papillary RCC, clear cell papillary RCC, unclassified RCC, chromophobe RCC, oncocytoma, and other rare renal tumors. Future studies should include other renal tumor types and histologic variants in matched cases for comparison.” (lines 279-283)

Reviewer #3 (Remarks to the Author):

The authors have done a good job revising the manuscript.

Comment 5: I have one remaining point, regarding the robustness of the number of bins. In their rebuttal, the authors say: "Empirically, the results are robust when the number of bins is set to around 10":

- Is this number chosen using a cross-validation approach? It needs to be done this way.
- Can you show a plot of AUC vs number of bins?

Response: *A five-fold cross-validation (CV) approach is used to decide the number of bins on dataset 1. The figure below shows the average AUC of the five-fold CV using a logistic regression classifier for each of the bin numbers ranging from 4 to 20. As we can see, the classification performance (AUC) is robust and stable in this bin size range and we set the number to 10, which corresponds to the highest point of AUC. This figure is added to supplementary information and cited in the manuscript (line 357).*